# ImmunizziAMO: A School-Based Field Trial to Teach New Generations the Importance of Vaccination through Games and to Fight Vaccine Hesitancy in Italy

**DOI:** 10.3390/vaccines8020280

**Published:** 2020-06-05

**Authors:** Giuseppe La Torre, Valeria D’Egidio, Cristina Sestili, Rosario Andrea Cocchiara, Sara Cianfanelli, Ornella Di Bella, Lorenza Lia, Barbara Dorelli, Vittoria Cammalleri, Insa Backhaus, Federica Pagano, Chiara Anguissola, Amelia Vitiello, Rita Carsetti, Alice Mannocci

**Affiliations:** 1Department of Public Health and Infectious Diseases, Sapienza University of Rome, 00185 Rome, Italy; valeria.degidio@uniroma1.it (V.D.); cristina.sestili@uniroma1.it (C.S.); rosario.cocchiara@uniroma1.it (R.A.C.); sara.cianfanelli@uniroma1.it (S.C.); ornella.dibella@uniroma1.it (O.D.B.); lorenza.lia@uniroma1.it (L.L.); barbara.dorelli@uniroma1.it (B.D.); vittoria.cammalleri@uniroma1.it (V.C.); insa.backhaus@uniroma1.it (I.B.); f.pagano@uniroma1.it (F.P.); alice.mannocci@uniroma1.it (A.M.); 2Explora il Museo dei Bambini di Roma, 00196 Rome, Italy; c.anguissola@mdbr.it; 3Comitato nazionale contro la meningite, 04100 Latina, Italy; avv.ameliavitiello@gmail.com; 4Immunological Diagnosis Unit, Bambino Gesù Children Hospital, 00165 Rome, Italy; rita.carsetti@opbg.net; 5Giochiamo Collaborative Group, Sapienza University of Rome, 00185 Rome, Italy; gigliolag020@gmail.com

**Keywords:** game, vaccination, field trial, school, primary prevention

## Abstract

Background: Vaccines simulate the first contact with infectious agents and evoke the immunological response without causing the disease and its complications. High rates of immunization among the population guarantee the interruption of the transmission chain of infectious diseases. Therefore, the population should be aware of the value of vaccination and motivated. In order to implement the spread of a correct culture about these issues, schools were recognized as a privileged operational setting. The aim of this project was to transmit knowledge and convey educational messages on the importance of vaccines, through the use of games, in elementary school children, their families and teachers. Materials and Methods: A field trial study was implemented between April and October 2019. Sample size calculations highlighted the need to recruit at least 136 students in the schools. The intervention involved 10 classes (five first grade and five s grade classes) and was structured in frontal teaching sessions and gaming sessions. Knowledge was assessed comparing the results of a questionnaire administered before and after the intervention. The questionnaires referred to the following items: dangerousness of bacteria and viruses; capability of defending from microorganisms; the role of antibodies; functioning of the vaccine in a child; type of disease for which a vaccine is efficacious; duration of a vaccine; mother- child transmission of antibodies; herd immunity. Results: 143 children participated in all the phases of the study. The comparison between the scores at the beginning and end of the intervention showed a significant increase in the knowledge about vaccines and immunity. The mean knowledge score arose from 3.52 (SD = 1.67) to 5.97 (SD = 1.81). Conclusions: This study suggests that the use of games in an elementary school effectively increase the knowledge related to the important topic of vaccination starting at childhood.

## 1. Introduction

Reaching high vaccination coverage is the main strategy for preventing vaccine preventable infectious diseases. The Italian Ministry of Health, in the National Plan for Elimination of Measles and Rubella (PNEMoRc) [1], recalls that it is necessary to interrupt the local transmission chain, in order to achieve elimination of endemic measles and rubella, and to reach this goal, vaccination coverage needs to be brought above 95% [2]. In Italy, in 2017 a measles outbreak occurred with over 4000 cases most of which were unvaccinated. In response to the emergency, 10 vaccinations (diphtheria, tetanus, pertussis, hepatitis B, poliovirus, *Haemophilus influenzae* type b, measles, mumps, rubella and chickenpox) become mandatory by law for children between the ages of 0 and 16 years [3]. A heated debate followed in civil society and politics, also triggered by the spreading of fake news about the lack of safety of the vaccines and a hypothetical link between vaccines and autism. As a result, a significant part of the general population, known as *no-vax,* started expressing concerns and negative attitudes toward vaccines.

The introduction of vaccines has led to a decline in preventable diseases; however, on the other side, a diminished sense of alarm and risk about infectious diseases among the population occurred. Vaccine Hesitancy, which is defined by the WHO as a “delay in acceptance or refusal of vaccines despite availability of vaccination services” [4] is still a real issue in Italy. Despite vaccination coverage increasing, among the 21 Italian regions only Tuscany exceeds 95% coverage for measles, rubella, and mumps, while for chickenpox only two regions (Basilicata and Puglia) have more than 90% coverage percentages [5].

The SAGE Working Group on Vaccine Hesitancy in 2015 suggested that to pursue the goal of defeating vaccine hesitancy, the WHO should engage allies from civil society organizations at global, regional and country level by integrating skills and knowledge of different professionals (e.g., sociologists, behavioral psychologists, anthropologists, experts in social marketing and communication, as well as specific disease experts) [6]. Considering civil society, children in the primary school might be a target of major interest. Vaccinations involve them directly and by conveying to children the principles of safety and importance of vaccinations it might be possible to reach parents, relatives, and teachers.

The school does not educate children and adolescents about vaccines and adults do not have a real understanding of the risks connected with vaccine-preventable diseases and the benefits of immunization for the health of the individual and the well-being of the whole community. Ensuring education and knowledge dissemination about vaccines among the younger generations through school-based programs might be a good strategy to strengthen parental acceptance and shape the future vaccine acceptance behavior of people who will soon become parents and adults [7].

This health promotion intervention called *“ImmunizziAMO”* is an innovative prevention model set in the primary school, based on the transmission of knowledge and on active learning achieved through the use of games. The initiative is part of a broad strand of studies carried out by La Torre’s working group, which are aimed to test game-based interventions as a means to increase children’s knowledge about preventable health issues. Previous studies were focused on modifiable risk factors of chronic non-communicable diseases, such as poor lifestyles (unhealthy eating, lack of physical activity, smoking habit, and alcohol consumption) [8,9,10]. Following the same methodology described in the above cited trials, *“ImmunizziAMO”* focuses on prevention of infectious diseases for which active immunoprophylaxis protocols are available.

The aim of this study is to teach the importance of vaccines and herd immunity starting from childhood. The project aims are as follows:to increase knowledge about communicable diseases and the concept of herd immunity;to make vaccination understood as a prevention tool;to change behaviors through reliable sources of information;to promote the achievement of high vaccination coverage.

To determine a real change in behavior, the traditional transmission of knowledge was supported by games, including card, board, and movement games. These are a growth tool with an established educational and pedagogical value, and the effectiveness of game theory is determined by scientific evidence that cognitive and behavioral aspects are reinforced with play [11].

## 2. Materials and Methods

The field trial was carried out between April 2019 and October 2019 at Podere Rosa and San Cleto schools in Rome, in the neighborhood of San Basilio, one of the most deprived areas in the metropolitan area of Rome [12]. Children were recruited from 10 classes, including 5 first grade and 5 s grade classes. We selected the schools on the basis of the lowest vaccination rate in the Local Health Unit in the municipal area of Rome. All the 1st and 2nd classes of the two schools were taken into account. Before starting the intervention, two preliminary meetings took place to present the project to teachers, and to children’s parents. After this, the trial with children started.

The study was structured in several stages, carried out on three different days, each one a few weeks apart from the others, with a separate and detailed schedule.

### 2.1. Day 1

In the first phase the project was presented to the classes (teachers and children together). Immediately afterwards, children were given a questionnaire, specially designed for primary school pupils, aimed at assessing their knowledge about infectious diseases, vaccinations, and herd immunity in the pre-intervention phase. We selected the items of the questionnaire on the basis of the educational contents (i.e., the video, the cards, and the board game) to be presented in the intervention.

The questionnaire included the following 8 questions:Are bacteria and viruses dangerous?How do you defend the human body from bacteria and viruses?What do antibodies do?What does the vaccine do to the child?Do you think the vaccine protects against disease?How long does the vaccine protect you?Does the mother pass antibodies to her child?Do you think vaccinated children protect unvaccinated children?

After the questionnaire was completed, students were given a lesson that explained the topics contained in the questionnaire: what are vaccines for, what they contain, how they act, what infectious diseases can be defeated with vaccines, and what is herd immunity. The lesson was held using a video as teaching support, specifically designed to stimulate children’s curiosity and promote the acquisition of notions without generating boredom and moments of distraction. The video, in the form of a cartoon, consisted of short scenes lasting up to 2–3 min depicting the concepts to be communicated. After each scene, the video was interrupted to allow the researchers to intervene, so that they could add explanations and stimulate dialogue in the classroom.

After watching the video, physical activities such as dodge ball game were undertaken. These activities were contextualized in that the children who hit the other players represented the microorganism. Unvaccinated children had to avoid the ball, while vaccinated individuals had the prerogative of protecting the susceptible ones. This game was functional in understanding herd immunity.

### 2.2. Day 2

During the second day, board and card games, specifically designed for the purpose of the intervention, were used for playing with the children.

The game sessions were coordinated by teachers, lasted about 60 min each, and the settings were the classrooms. Children were grouped in small units (six to ten persons each), and a total of three games (board and card games) was carried out.

The winning playing cards depicted children with shields (suggesting the value of vaccination), while the loser cards had figures of microorganisms on them (Figure 1). These card games followed the rules of games belonging to the Italian tradition. For example, “Zompa Vaccino” which means “Jumping vaccine”, was based on the scheme of a popular card game, named “Salta cavallo” (Jumping horse).

Another card game delivered was inspired by a further traditional Italian card game, ‘‘Tappo.’’

A board game called *The Vaccine Goose* was also used (Figure 2), revisiting the well-known Game of the goose. In this revised version, the winning boxes were represented by vaccinated individuals and allowed the gamer to reach the finish line earlier; conversely, cells with pathogen microorganisms, decelerated the path of the participant.

### 2.3. Day 3

The last educational phase took place at Explora, the Children’s Museum in Rome. Each activity lasted 15 min and consisted of interactive games that strengthened issues already addressed in the previous intervention sessions. The workshops were as follows:“I defend myself” to find out how the immune memory generated by the vaccine works. A puzzle game explaining how the antibody binds to the antigen.“If I protect myself, I protect you” about the concept of herd immunity consisting of a game of stamps and inks to create a poster to take to school.“I wash my hands” on the importance of washing hands. It was explained how long do germs and bacteria live? The concept of contamination was shown with Petri dishes and microscopes“I present to you the vaccine” a fun quiz to learn together how to defend ourselves by developing defenses immunity. At the end of the game, a certificate was given to everyone.

Finally, the intervention was concluded by administering the children with the same questionnaire proposed at the beginning of the trial, to detect any difference between pre-and post- intervention knowledge. In addition, each child was asked to create a drawing with what most impressed him about this experience and to express an opinion or a feeling about what he or she learned.

### 2.4. Sample Size and Statistical Analysis

Using the software Epicalc2000, we assessed the sample size using the following parameters:Mean score pre-intervention: 4.00 pointsMean score pre-intervention: 5.00 pointsSD: 2.00 pointsSignificance: 0.05Power: 80%

According to these parameters, we had the need to recruit 124 students. Moreover, we increased the total number by 10%, considering possible loss to follow up, recruiting at least 136 students in the two schools.

A pilot study involving 26 students revealed a Spearman correlation coefficient for test-retest reliability of 0.873 (*p* = 0.001) and Chronbach alpha of 0.702.

To perform statistical analysis and compare results gained by the students, a scoring system was applied. The score was given by the sum of the correct answers. Scores indicated as T0 referred to baseline results, and to T1, if referring to post-intervention results.

Then, the primary outcome was expressed as the difference between T1 scores and the related T0 scores. This difference was calculated and named as the Delta score, to underline changes in time.

The statistical analysis was conducted using SPSS, release 25.0.

Differences in the scores were tested using the T-test for paired data (differences in the score pre- post) and T-test for independent samples (differences between groups).

The statistical significance was set at *p* < 0.05.

## 3. Results

### 3.1. Descriptive Analysis of the Sample

A total of 171 children took part in the pre-intervention questionnaire about knowledge, with a median age of 7 years (minimum 6, maximum 8). Most of the participants were males (90) representing the 52.6% of the sample population. Only two children did not take part in the intervention, due to the lack of parental consent.

A total of 28 children (16.4%) did not complete the intervention for reasons mainly related to absence from school during the post-intervention questionnaire administration day. There were no differences in the knowledge score between those who completed the intervention and those who did not, either by gender (*p* = 0.246), by class (*p* = 0.733), or by the school complex which was considered (*p* = 0.670), or by age (*p* = 0.409), or by mean score of pre-intervention knowledge (*p* = 0.317).

### 3.2. Analysis of Knowledge

Table 1 shows the frequencies of correct answers to each item of the knowledge questionnaire before and after the intervention.

In the pre-intervention phase, the mean knowledge score was 3.48 (SD = 1.65), with a Gaussian response distribution (Figure 3).

143 children filled the post-intervention questionnaire: 66 females (46.2%) and 77 males (53.8%). The mean knowledge score was 5.97 (SD = 1.81), with a distribution of scores with a marked left oriented tail (Figure 3).

The T-test for matched data shows that the knowledge score considerably increased (*p* < 0.001). Among the 143 children that filled both pre- and post-intervention questionnaires, the mean knowledge score rose from 3.52 (SD = 1.67) to 5.97 (SD = 1.81).

No significant differences were observed in the changing of knowledge scores, either by class, school complex, or gender (Table 2).

## 4. Discussion

The significant difference in knowledge score recorded before and after the intervention confirms that learning through play can be a way forward to broaden the spectrum of health topics and to educate young people.

A few studies from the scientific literature were aimed at experimenting innovative strategies in promoting children’s health. In particular, several trials addressed the entire population of a particular geographic area [13,14] and young generations were rarely targeted in the first person, as these studies often targeted pre-teens or teenager populations. Furthermore, many studies were performed aiming at improving parental knowledge and acceptance towards vaccines [15,16,17,18,19,20].

There are examples of trials that involved children directly and at the same time were based on innovative strategies of knowledge transmission, as sport or recreational activities, and measured the efficacy of intervention by recording the increase in knowledge increase [21]. These studies confirm several findings resulting from our field trial:game-based health promotion interventions are effective with young children, as well as adolescents and young adults;game-based health promotion interventions can be tailored to educate about infectious diseases;positive effects on mental health and social behavior of children to be taken into account;long-and short-term interventions may gain similar efficacy.

A relevant aspect is the importance of stimulating community engagement as a part of a broader approach to community health issues. It has been described that pre-existing community accountability dynamics support this kind of intervention within communities [14]. Consequently, it might be relevant for the effectiveness of these interventions that community leaders (teachers) are engaged alongside health professionals and prevention experts. This “alliance” can strengthen and facilitate the intervention of researchers and confer accountability from children and their parents. It is important to underline that the study provides no additional knowledge about viruses but focuses on educating young children to understand the value of vaccination, and consequently the data obtained nevertheless pertains more to the science of education than to the science of vaccines.

One possible limitation of our trial is the small sample size and the fact that the intervention took place, like the previous ones, only in two schools. For this reason, it would be desirable to carry out new health education interventions in a greater number of school complexes. It would be desirable to conduct the trial on a larger scale, at the regional level for example. Another possible limitation could be represented by the low socio-economic status of the neighbourhood chosen for the trial. However, since the results are promising we could imagine that similar results would be obtained in less deprived areas.

It would also be appropriate to propose the intervention in different settings, such as hospitals. Pediatric patients often face health issues without having a full understanding of them. By this kind of intervention, in fact, children would benefit from playful educational activities, gaining improvement in both learning and emotional aspects, as well as social behavior.

## 5. Conclusions

The *ImmuniziAMO* project, carried out in 10 classes of two schools in Rome, demonstrated that games are an effective strategy for teaching children the function and value of vaccination.

Students’ level of knowledge considerably increased in the post-intervention phase. Students understood the utility of vaccination as a prevention tool, and they acquired the concept of herd immunity. The children in their drawings showed appreciation for the game sessions and for the pictures printed on the cards.

This field trial successfully pursued the intent of educating on a subject of substantial importance, on which no reliable and accurate information is available for children of early school age. In addition, our trial reached a young population, which turned out to be incredibly receptive to the message of primary prevention.

The project was based on the transmission of knowledge and active learning through the use of a game-based strategy. In our view, what was carried out here in Rome could represent an example worth following by many more.

The satisfactory results that were obtained indicate that some of the communication tools necessary to address this particular population are as follows: the administration of little but well- stated information alternating with short recreational moments; the use of popular games, selected to fit the matter and tailored to attract the attention of children; the intention to stimulate children’s curiosity towards subjects that are also often challenging for adults.

## Figures and Tables

**Figure 1 vaccines-08-00280-f001:**
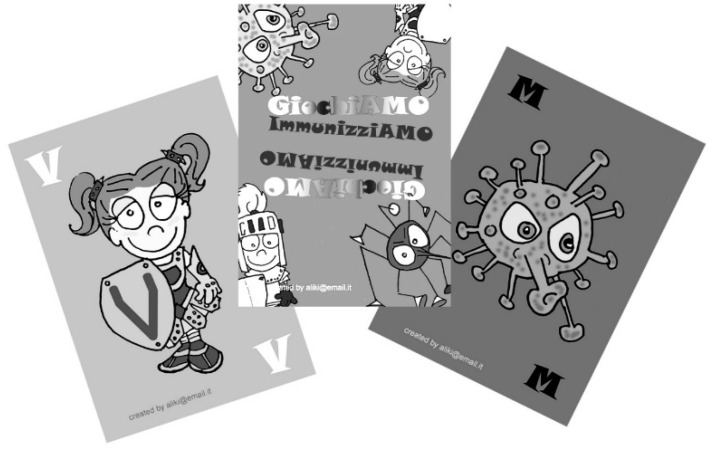
Examples of cards used in the Immunizziamo project.

**Figure 2 vaccines-08-00280-f002:**
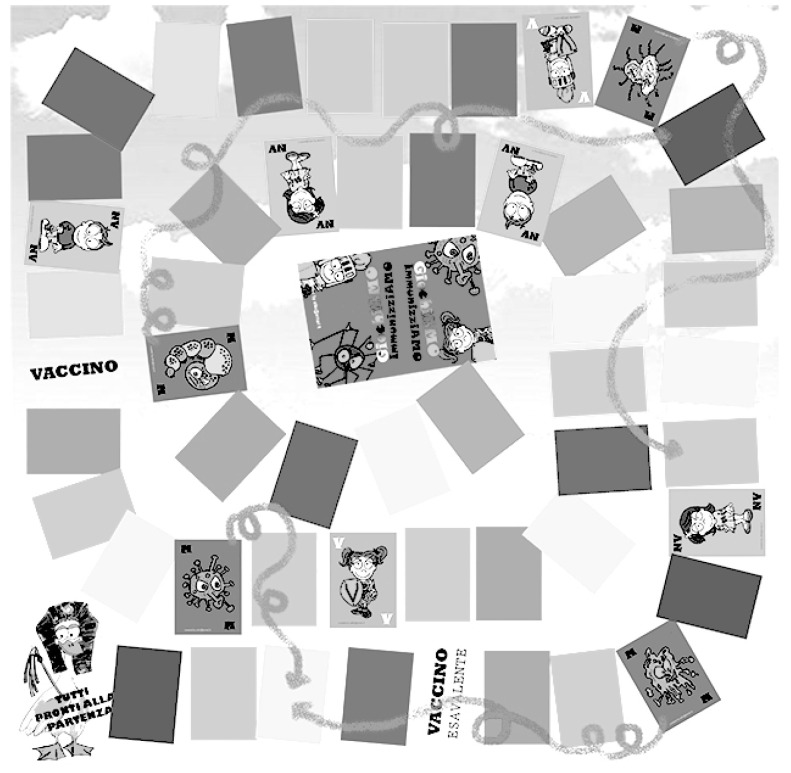
The Vaccine Goose board game.

**Figure 3 vaccines-08-00280-f003:**
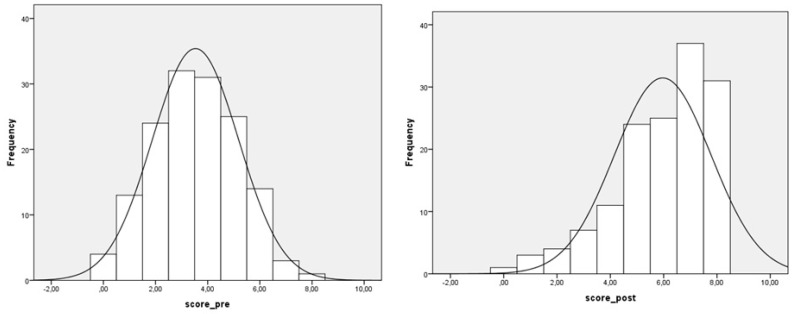
Pre-intervention and post-intervention knowledge score frequency distributions.

**Table 1 vaccines-08-00280-t001:** Correct answers (proportion) to each of the 8 questions of the questionnaire (pre-intervention compared to post-intervention).

Question	Pre-Intervention	Post-Intervention
1	83 (48.5%)	115 (80.3%)
2	57 (33.3%)	91 (63.9%)
3	75 (43.9%)	108 (75.5%)
4	102 (59.6%)	114 (79.6%)
5	60 (35.1%)	75 (52.7%)
6	94 (55%)	105 (73.3%)
7	57 (33.3%)	114 (79.6%)
8	67 (39.2%)	130 (91%)

**Table 2 vaccines-08-00280-t002:** Differences in delta knowledge scores, calculated by T-test for matched data.

Group	Mean	SD	*p* Value
First classes	2.55	2.23	0.619
Second classes	2.36	2.12	
Podere Rosa school	2.41	2.18	0.666
San Cleto school	2.61	2.17	
Females	2.58	2.33	0.740
Males	2.32	2.06

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
