# Peer review of "ImmunizziAMO: A School-Based Field Trial to Teach New Generations the Importance of Vaccination through Games and to Fight Vaccine Hesitancy in Italy"

_vaccines, 2020, doi:10.3390/vaccines8020280_

Round 1

Reviewer 1 Report

This study is interesting and clearly presented. A little bit of polishing of englsih usage would however be warranted. The study provides no additional knowledge about viruses but focuses on educating young children to understand the value of vaccination. I believe the goals pursued are worthwhile and well performed. The data obtained nevertheless pertains more to the science of education than to the science of vaccines.   

Author Response

Reviewer 1

This study is interesting and clearly presented. A little bit of polishing of englsih usage would however be warranted. The study provides no additional knowledge about viruses but focuses on educating young children to understand the value of vaccination. I believe the goals pursued are worthwhile and well performed. The data obtained nevertheless pertains more to the science of education than to the science of vaccines.   

Answer to the reviewer 1

We thank the reviewer for her/his commet. We completely agree with what the reviewer says “The study provides no additional knowledge about viruses but focuses on educating young children to understand the value of vaccination”, and that “The data obtained nevertheless pertains more to the science of education than to the science of vaccines”. We added a paragraph on these issues in the Discussion section.

Reviewer 2 Report

This is an interesting paper presenting the education program “ImmunizziAMO: a school-based field trial to teach new generations the importance of vaccination through games and to fight vaccine hesitancy in Italy” that was set up and carried out in some Italian primary schools.

The topic of this paper deals with an actual subject in Italy and maybe in other countries and proposes a school program/play directed to new generations as a tool to make children aware of the role played by vaccination in preventing diseases.

The new generations’ education could also be the way to forward the message to parents and relatives in a context of poor understanding of vaccination strategy for human health.

Statistical analysis of the results obtained have also been performed.

As remarked by the Authors, a limit of this paper is the limited number of schools and children involved in this program. However, what has been done in Rome could represent an example to be followed by many others.

Minor comments are included in the text.

English language need a deep revision by an English native speaker.

Author Response

Reviewer 2

This is an interesting paper presenting the education program “ImmunizziAMO: a school-based field trial to teach new generations the importance of vaccination through games and to fight vaccine hesitancy in Italy” that was set up and carried out in some Italian primary schools.

The topic of this paper deals with an actual subject in Italy and maybe in other countries and proposes a school program/play directed to new generations as a tool to make children aware of the role played by vaccination in preventing diseases.

The new generations’ education could also be the way to forward the message to parents and relatives in a context of poor understanding of vaccination strategy for human health.

Statistical analysis of the results obtained have also been performed.

As remarked by the Authors, a limit of this paper is the limited number of schools and children involved in this program. However, what has been done in Rome could represent an example to be followed by many others.

Minor comments are included in the text.

English language need a deep revision by an English native speaker.

Answer to the reviewer 2

We thank the reviewer for her/his comment. We completely agree with what the reviewer says “what has been done in Rome could represent an example to be followed by many others”.

We added a paragraph on these issue in the Discussion section.

Moreover, a linguistic revision has been performed by an English native speaker

Reviewer 3 Report

Thank you for giving me the opportunity to review the article. The authors conducted a study on the school-based field trial to teach new generations the importance of vaccination through games and to fight vaccine hesitancy in Italy. The topic is socially important, but crucial methodological concerns exist. I listed the comments below.

Comments:

Article Type:

  1. “Field trial” is not a type of article accepted in the journal. The authors should check the Instructions for Authors, and the type should be selected from them.

Abstract:

  1. The sample size should be mentioned in the Materials and Methods section.
  2. A brief explanation of questionnaire should be added.

Materials and Methods:

  1. How did the authors select the schools and classes? This should be described. The recruiting methods also should be added.
  2. How did the authors calculate the number of study participants of this study?
  3. How did the authors make the test used in this study? Can the test evaluate the knowledge appropriately? This should be justified.
  4. The authors should clearly show the primary and secondary outcomes of this study.
  5. The statistical analysis section should be added.

Results:

  1. The socioeconomic backgrounds of study participants should be presented.

Discussion:

  1. The authors should think about other limitations of this study to conduct future investigations.

Author Response

Reviewer 3

Thank you for giving me the opportunity to review the article. The authors conducted a study on the school-based field trial to teach new generations the importance of vaccination through games and to fight vaccine hesitancy in Italy. The topic is socially important, but crucial methodological concerns exist. I listed the comments below.

Answer:

we really thanks reviewer 3 for her/his valuable comments and suggestions. We hope now the manuscript has been improved in this revised version

Comments:

Article Type:

  1. “Field trial” is not a type of article accepted in the journal. The authors should check the Instructions for Authors, and the type should be selected from them.

Answer:

The Instructions for authors (see below) does not refer to the study design

Articles: Original research manuscripts. The journal considers all original research manuscripts provided that the work reports scientifically sound experiments and provides a substantial amount of new information. Authors should not unnecessarily divide their work into several related manuscripts, although Short Communications of preliminary, but significant, results will be considered. Quality and impact of the study will be considered during peer review.

However, the wording “Field trial” is widely accepted in the Education field.

Moreover, the Journal Vaccines published in December 2019 a paper entitled “Impact of Communicative and Informative Strategies on Influenza Vaccination Adherence and Absenteeism from Work of Health Care Professionals Working at the University Hospital of Palermo, Italy: A Quasi-Experimental Field Trial on Twelve Influenza Seasons”.  So we would like to leave the title as it is.

Abstract:

  1. The sample size should be mentioned in the Materials and Methods section.

Answer:

We added that we need to recruit at least 136 students in the two schools.

  1. A brief explanation of questionnaire should be added.

Answer:

in the abstract now we add a brief explanation of the questionnaire:

“pre and post-intervention questionnaires refers to the following items:  dangerousness of bacteria and viruses; capability of defending from microorganisms; role of the antibodies; functioning of the vaccine in a child; type of disease for which a vaccine is efficacious; lasting of a vaccine; mother-child transmission of antibodies; heard immunity”

Materials and Methods:

  1. How did the authors select the schools and classes? This should be described. The recruiting methods also should be added.

Answer:

We selected the schools on the basis of the lowest vaccination rate in the Local health Unit in the municipal area of Rome.

All the 1st and 2nd classes of the two schools were taken into account

  1. How did the authors calculate the number of study participants of this study?

Answer:

Using the software Epicalc2000, we used the following parameters

Sample - Size - Two means

20:22:55, 24/01/2019

Mean 1                             :       4,00     

Mean 2                             :       5,00     

SD                                         :       2,00     

Significance                     :       0,05     

Power                               :      80%       

Sample size                      :      62         (each group)

                                               :     124         (overall)

So, we had the need to recruit 124 students. Moreover, we increased the total number by 10%, considering possible lost to follow up, recruiting at least 136 students in the two schools.

  1. How did the authors make the test used in this study? Can the test evaluate the knowledge appropriately? This should be justified.

Answer:

we selected the items of the questionnaire on the basis of the educational contents (i.e., the video, the cards and board game).

  1. The authors should clearly show the primary and secondary outcomes of this study.

Answer:

We had only a primary outcome in this study and this has been highlighted in the methods section

  1. The statistical analysis section should be added.

Answer:

A statistical paragraph has been added

Round 2

Reviewer 3 Report

Thank you for giving me the opportunity to review the revised version of this article. The authors conducted a study on the school-based field trial to teach new generations the importance of vaccination through games and to fight vaccine hesitancy in Italy. The topic is socially important, but crucial methodological concerns exist. I listed the comments below.

AC: Additional Comments

Comments:

Article Type:

  1. “Field trial” is not a type of article accepted in the journal. The authors should check the Instructions for Authors, and the type should be selected from them.

AC: The authors mentioned about an article published in the journal Vaccines. The article published as the category of “Article”. If the authors want to publish the article in the journal, you should follow the instruction for authors.

The authors should check the PDF version of the following article.

Ref: https://www.mdpi.com/2076-393X/8/1/5

Materials and Methods:

  1. How did the authors make the test used in this study? Can the test evaluate the knowledge appropriately? This should be justified.

AC: I understood that the authors selected the items of the questionnaire on the basis of the educational contents. IT means that the test did not validate before using in this study. Therefore, the authors should discuss about it in the Discussion section.

  1. AC: The authors should provide the questions used in this study. The reviewer and potential readers cannot know about the contents of the questions (mentioned in the Table 1 as “1” to “9”).

Results:

  1. The socioeconomic backgrounds of study participants should be presented.

AC: The reviewer cannot get any answer to this point.

  1. AC The authors should show the number of correct answers with its percentage (i.e. n (00.00%)) in the Table 1.

Discussion:

  1. The authors should think about other limitations of this study to conduct future investigations.

AC: The reviewer cannot get any answer to this point.

Author Response

AC: Additional Comments

Comments:

Article Type:

  1. “Field trial” is not a type of article accepted in the journal. The authors should check the Instructions for Authors, and the type should be selected from them.

AC: The authors mentioned about an article published in the journal Vaccines. The article published as the category of “Article”. If the authors want to publish the article in the journal, you should follow the instruction for authors.

The authors should check the PDF version of the following article.

Ref: https://www.mdpi.com/2076-393X/8/1/5

Answer

There was a misunderstanding on this point, since we understand that the reviewer was referring to the title. Now we remove “Field trial” from the space over the title and replace with “Article”

Materials and Methods:

  1. How did the authors make the test used in this study? Can the test evaluate the knowledge appropriately? This should be justified.

AC: I understood that the authors selected the items of the questionnaire on the basis of the educational contents. IT means that the test did not validate before using in this study. Therefore, the authors should discuss about it in the Discussion section.

Answer

A pilot study involving 26 students revealed a Spearman correlation coefficient for test-retest reliability of 0.873 (p=0.001) and  Chronbach alpha of 0.702. This was added at page 6, lines 207-208.

  1. AC: The authors should provide the questions used in this study. The reviewer and potential readers cannot know about the contents of the questions (mentioned in the Table 1 as “1” to “9”).

Answer

The questions used in the study are already described in the manuscript at page 4 (lines 120-127).

Results:

  1. The socioeconomic backgrounds of study participants should be presented.

AC: The reviewer cannot get any answer to this point.

Answer

The socio-economic status of the study participants was not assessed due to the young age of the respondents (6-8 years). However, we introduced a statement at the beginning of the Methods section on the neighborhood of San Basilio, one of the most deprived area in the metropolitan area of Rome and added the following reference:

Celata F, Lucciarini S. Atlante delle disuguaglianze a Roma [Atlas of inequalities in Rome]. Camera di Commercio Industria Artigianato e Agricoltura di Roma. Rome, October 2016. Available at https://web.uniroma1.it/memotef/sites/default/files/Atlante_Camcom_2016_compresso.pdf

AC The authors should show the number of correct answers with its percentage (i.e. n (00.00%)) in the Table 1.

Answer

We added the absolute number in Table 1 as suggested

Discussion:

  1. The authors should think about other limitations of this study to conduct future investigations.

AC: The reviewer cannot get any answer to this point.

Answer

We added another possible limitation. The following statement was included in the discussion section:

Another possible limitation could be represented by the low socio-economic status of the neighbourhood chosen for the trial. However, since the results are promising we could imagine that similar results could be obtained in less deprived areas.

Round 3

Reviewer 3 Report

Thank you for giving me the opportunity to review the revised version of this article. The authors revised the manuscript mostly appropriately. However, the reviewer would like the authors to format the Figure 3 for publication before acceptance. The current figure my be pasted from the output which obtained from the statistical software.

Author Response

We replace the figure 3 made using a past-copy from the SPSS software with a figure made with a jpeg format.

Many thanks

Giuseppe La Torre